# Preparation and Characterization of Spherical Amorphous Solid Dispersion with Amphotericin B

**DOI:** 10.3390/pharmaceutics10040235

**Published:** 2018-11-16

**Authors:** Lyes Mehenni, Malika Lahiani-Skiba, Guy Ladam, François Hallouard, Mohamed Skiba

**Affiliations:** 1UFR of Health, Laboratory of Pharmaceutical & Biopharmaceutical technology, Inserm U1239, UNIROUEN, Normandy University, 76183 Rouen CEDEX, France; lyesmehenni@yahoo.fr (L.M.); malika.skiba@univ-rouen.fr (M.L.-S.); francois.hallouard@gmail.com (F.H.); 2PBS (UMR 6270), CNRS, UNIROUEN, Normandy University, 27000 Evreux, France; guy.ladam@univ-rouen.fr

**Keywords:** amorphous solid dispersions, amphotericin B, cyclodextrins polymers, spray-drying

## Abstract

In the present study, new polymer microspheres of amphotericin B (AmB) were prepared by a spray drying technique using cyclodextrin polymers (Poly-CD) to improve the solubility and dissolution of AmB, to prevent in vivo toxic AmB aggregations. Formulations were characterized through scanning electron microscopy (SEM), Fourier transform infrared spectroscopy (FT-IR), differential scanning calorimetry (DSC), thermal analysis, Raman spectroscopy, particle size, drug purity test and in vitro release studies. The analysis indicated that the chemical structure of AmB remained unchanged in the amorphous solid dispersion, but the structure was changed from crystalline to amorphous. AmB was completely release from such optimized formulations in dissolution media in 40 min. This work may contribute to a new generation of spherical amorphous solid dispersion using a cyclodextrin polymer, which has implications for the possibility of drug development for oral utilization or as powder aerosols for pulmonary administration.

## 1. Introduction

Amphotericin B [AmB] is considered as an old drug with antifungal properties. This active pharmaceutical ingredient was extracted from *Streptomyces nodosus* and its chemical synthesis was discovered in 1970 [1]. In medicine, AmB is used for the treatment of deep or systemic fungal development induced by an immunosuppressive state of patient which is one of the major causes of mortality [2].

AmB belongs to class IV in the Biopharmaceutical Classification System, presenting low solubility (about 1 µg/mL in water) and low membrane permeability [3]. This low solubility is due to the self-association of AmB molecules above a critical micellar concentration at about 0.2 mg/mL [4]. The long polyene chain measuring 21 Å is responsible for the high absorption between 300 and 450 nm allowing its dosage using an ultra-violet [UV] spectrophotometer.

Recently, several formulations were described to enhance AmB solubility, such as amorphous drug nanoparticles [5], carbon nanotubes [6], chitosan microparticles [7], cubosomalnanoparticles [8], nano-emulsions [9], gelatin nanoparticles [10], polymeric nanoparticles [11], solid lipid nanoparticles [12] or liposomes [13].

The marketed available preparations of AmB are AmBisome^®^ and Fungizone^®^. These preparations are indicated for the intravenous route and Fungizone is also indicated per os. However, the high interaction of AmB molecules with plasma proteins induces nephrotoxicity [14]. Hepatotoxicity was also described at therapeutic doses corroborating with aggregated states of AmB: dimeric or poly-aggregated forms. These aggregated states of AmB depend on different formulation factors such as pH and the presence of some excipients [15,16].

To prevent in vivo toxic aggregated forms, the use of cyclodextrins [CD] or derivatives seems interesting. Indeed, CDs have the ability to form inclusion complexes in their inner cavity with host molecules preventing aggregation. CD and their derivatives are also known to be efficient solubilizing agents [17]. CD-based polymers [Poly-CD] were developed in order to increase the solubility of natural CD or derivatives [18]. In addition, natural CD or Poly-CD as solubilizing agents are considered as safe for a pharmaceutical use because of their non-toxic composition, mainly based on glucose units. 

A new approach is essential to implement the large capacity of this antifungal drug. To avoid the conventional routes of administration and to enhance the local drug bioavailability, another alternative is found by inhalation AmB particles to target local action in the lungs. A recent study has showed interest in using a pulmonary route for administering microparticles and lipid complexes [19]. The results showed a good pulmonary deposition and a lower rate of active substance in the kidney leading to a reduction of drug nephrotoxicity and an improvement of fungal activity, approximately 1.5-times better than the marketed intravenous preparations. Pharmacokinetic and organ distribution studies showed that the pulmonary administration may be favorable for a good efficiency by local action [19].

The aim of this work is to design and evaluate in vitro original formulations of AmB based on cyclodextrin or derivatives for lung administration. 

To reach a targeted release, the design of amorphous solid dispersions using Poly-CD will be studied. 

## 2. Materials and Methods

### 2.1. Chemicals and Materials

Amphotericin B as active pharmaceutical ingredient was purchased by Quimdis (Levallois-Perret, France). Natural α-cyclodextrins (α-CD), β-cyclodextrins (β-CD) and γ-cyclodextrins (γ-CD) were provided by Wacker (München, Germany) and permethylated β-CD [PMβ-CD] by Roquette (Lestrem, France). Homopolymers, terpolymers and tetrapolymers of CD, were acquired from start-up In-Cyclo (Rouen, France). The surfactant Sodium Lauryl Sulfate (SLS) was obtained from VWR (Radnor, PA, USA). All other chemicals and the reagents were commercially available products of analytical grade.

### 2.2. Amphotericin B Determination-HPLC/UV Method

AmB was analyzed, isolated and quantified using an HP 1100 series chromatographic system (Agilent Technologies, Santa Clara, CA, USA) equipped with a binary pump, an auto sampler and a diode array detector with a detection wavelength set at 405 nm. The separations were performed on a base-deactivated reversed phase Purospher^®^ Star RP-C_18_ column (Merck Chemicals, Belgium). The elution was realized by a mobile phase composed of MeOH/Acetonitrile /EDTA 0.25 mM buffer (40:30:30) (*v*/*v*/*v*), running at a flow rate of 1 mL/min. The injected volume was 10 μL and the column temperature was set at 25 °C during the analysis time. The LC-method was used for the solubility and lyodisponibility studies of the pure drug and the different matrix loaded with AmB.

### 2.3. Synthesis of Homopolymers, Terpolymers and Tetrapolymers of CD

Cyclodextrin polymers were synthesized by using a direct melt polycondensation process according to the method reported by Skiba [18]. Briefly, a mixture of known amount (*w*/*w*) of cyclodextrins (α, β, γ or per-methyl β), citric acid and sodium phosphate dibasic was transferred into a reactor which was maintained at temperature ranging between 140 °C and 150 °C for fixed time. The obtained solid form was dissolved in water and dialyzed using polyether sulfate membrane filter with molecular weight cut off of 10,000 Da. After the dialysis, the resulted solution was spray-dried with a Mini Spray Dryer B-290^®^ (BÜCHI, Flawil, Switzerland). 

### 2.4. Phase Solubility Studies

The experiments were in accordance to Higuchi & Connors [20] to evaluate the interactions between AmB and the cyclodextrin entity in terms of solubility. The different Poly-CD: Poly γ-CD (homopolymers), Poly αγ-CD and Poly βγ-CD (terpolymers) and Poly αβγ-CD (tetrapolymers) were solubilized in flask that containing 5 mL of distilled water. An excess amount of AmB was added in the saturation limit. The yellow-orange suspensions were magnetically stirred at 600 rpm at *T* = 37 ± 2 °C for three days and centrifuged at 10,000 rpm by Eppendorf 5800 Centrifuge (Hamburg, Germany). Finally, the supernatant fluid was filtered (0.2 µm) and analyzed by HPLC. The concentration range was between 25 and 1000 mg/mL for Poly-CD in accordance to their higher limit solubility which fixed at 1.2 g/mL [17].

### 2.5. Preparation of Amorphous Solid Dispersions

A Büchi 290 nozzle type mini spray-dryer (Flawil, Switzerland) was used for the preparation of AmB-loaded spherical amorphous solid dispersions. The cyclodextrin polymers used as homopolymers poly γ-CD, terpolymers poly αγ-CD and poly βγ-CD and tetrapolymer αβγ-CD are retained because of their greater ability to solubilize AmB and based on the results of phase solubility studies. The conditions of spray-drying were determinate by an optimization within the laboratory.

Amorphous solid dispersions were prepared with Poly-CD dissolved in 270 mL of deionized water (DW). The AmB powder was solubilized in 30 mL of methanol and sonicated for 10 min. The yellow and cloudy solution of AmB was added in polymer aqueous solution and stirred mechanically for 15 min to allow a better dispersion of AmB in the mixture. The amorphous solid dispersions with AmB content (approximately 1%, 7%, 15%) were prepared by spray-drying. The same method was used to prepare inclusion complexes based natural γ-CD. 

The amorphous solid dispersions were prepared from a lower AmB loading concentration than the equilibrium solubility of AmB in cyclodextrin polymers. The solutions were spray-dried with solution feed rate of 4 mL/min and outlet temperature of 130 °C. The inlet temperature was observed in the domain of 75–80 °C. The pressure of the aspirator filter vessel was maintained between *P* values of −50 and −40 mbar.

The formed solution or suspension was left under magnetic stirring throughout the process of atomization by spray-drying. A homogeneous powder was obtained at the end of the process.

### 2.6. Drug Loading and Entrapment Efficiency

The dosage of AmB in the different formulations was carried out by HPLC to estimate the amount present in each powder and to determine the efficiency of the atomization process. Besides, the homogeneity of AmB distribution in the matrix was verified by a triplicate analysis of the powder samples (*n* = 3). An equivalent of 50 mg of AmB were solubilized in 5 mL DMSO and completed to 100 mL by a mixture contained (H_2_O:methanol) (80:20) (*v*/*v*). The solution was sonicated and filtered through 0.45 μm PVDF, diluted and analyzed by HPLC. 

### 2.7. Evaluation of Aggregated States of AmB by Spectrophotometry UV

An equivalent of 1 mg of AmB was prepared from the amorphous solid dispersions (Poly γ-CD:AmB) in distilled water to obtain a concentration of 10 µg/mL, and analyzed by UV-1600PC UV/vis spectrophotometer (VWR, Radnor, PA, USA) using a quartz cell in the range of 300–450 nm. The same concentration of cyclodextrin polymers (Poly γ-CD) without AmB was used as blank. The UV spectrum of AmB pure drug, and AmB in polymeric matrix were compared with literature and marketed formulations.

### 2.8. Differential Scanning Calorimetry Property

Differential scanning calorimetry (DSC) was performed using DSC6^®^ (Perkin-Elmer, Waltham, MA, USA). An empty aluminum pan was used to calibrate the apparatus. Samples (containing 5 mg of amphotericin B) were accurately weighed into aluminum pans and sealed. Sample thermograms were obtained at a scanning rate of 10 °C/min conducted over a temperature range of 30–250 °C under a nitrogen gas (30 mL/min). The apparatus was controlled by the Pyris^®^ software (Perkin-Elmer, Waltham, MA, USA). Measurements were performed in triplicate.

### 2.9. Thermal Analysis

Mass losses were recorded with 4000^®^ (Perkin-Elmer, Waltham, MA, USA) on 5 mg samples in open pans at the heating rate of 10 °C in the 30–700 °C temperature range under a nitrogen gas flow (20 mL/min). All measurements (AmB pure drug, Poly-CD, Physical mixture and amorphous solid dispersions) were performed in triplicate.

### 2.10. Fourier Transformed Infrared Spectroscopy (FT-IR)

The interaction of AmB with natural CD and Poly-CD in the solid state from dried samples were studied by infrared spectroscopy. FT-IR spectra were recorded on an IR spectrophotometer from PerkinElmer equipped with an ATR, in the range between 4000 cm^−1^ and 700 cm^−1^ at a resolution of 8 cm^−1^ and with 20 scans. All measurements were performed in triplicate.

### 2.11. Raman Spectroscopy Property

Raman analyses were carried out by using a confocal Raman microscope (Lab Ram HR by Jobin-Yvon Horiba, Kisshoin, Minami-ku Kyoto 601-8510, Japan). The excitation of Raman scattering was operated by a He-Ne laser at a wavelength of 632.8 nm. The spectral resolution used was 2 cm^−1^. Raman spectra of the crystals with good signal-to-noise ratio was obtained with integration times from 30 to 90 s and the duration of the data collection was adjusted in order to minimize the background signal. The selected spectral lateral resolution was 4 cm^−1^ for a spatial lateral resolution better than 2 mm. All experiments were carried out at room temperature.

### 2.12. Particle Size Analysis

The measurement of particle size was performed using Malvern Mastersizer (Malvern Hydro 2000S, Grovewood Road Malvern WR14 1XZ, United Kingdom). Heptane was used as a dispersant (refractive index of 1.385–1389 and polarity index of zero) [21]. Each sample was dispersed in heptane and stirred at 2000 rpm in order to reduce the interparticle aggregation. The obscuration range was maintained between 10–20%. The average particle sizes were measured after performing the experiment for each batch in triplicate.

### 2.13. Scanning Electron Microscopy (SEM)

A Scanning electron microscopy (model JEOL JCM-5000, Musashino, Akishima, Tokyo 196-8558, Japan), NeoScope instrument was used for the study at an accelerated voltage between 10 and 15 kV. Powder samples were stuck on SEM stub with conductive adhesive tape and coated with gold to reduce electric charges induced during analysis with a NeoCoater MP-19020NCTR.

### 2.14. In Vitro Dissolution

In vitro dissolution studies of pure drug and solid dispersions prepared by the different methods were evaluated using Vankel VK 7000^®^ dissolution system (Varian Inc., Palo Alto, CA, USA). The rotation speed was 75 rpm and the temperature were adjusted at 37 °C ± 0.5 °C. 

The drugs prepared by spray-drying were tested in two different media with 0.10% (*m*/*v*) of SLS. The first was an acid medium at pH = 1.2 for the prediction of release capability in vivo using simulated gastric medium. The second was at pH = 7.4 having a similarity with simulated body fluid [22]. The composition of both dissolution media was:

Dissolution medium [1]: 7 mL of HCl (37%) was added in a 1-L flask containing 500 mL of water. 2 g of NaCl was added and the volume was completed to 1000 mL with H_2_O and pH adjusted to a value of 1.2 with HCl 0.1 M.

Dissolution medium [2]: the components described in Table 1 were simultaneous solubilized in 500 mL of H_2_O, the volume was completed to 1000 mL with H_2_O and pH adjusted to a value of 7.4 with HCl 0.1 M. The equivalent of 5 mg of AmB was predetermined and dissolved into vessels containing 250 mL of dissolution medium in compliance with sink conditions. SLS was selected as detergent for the dissolution test because of his strength ionic power in saline solution.

In all experiments, 2 mL of dissolution sample was withdrawn at 5, 10, 20, 30, 45, 60 and 90 min and replaced with an equal volume of the fresh medium supplemented with 0.1% or 0.02% SLS to maintain a constant total volume. Samples were filtered through 0.20 µm and assayed for drug content at 405 nm by UV spectrophotometry. The dissolution tests were conducted in triplicate (*n* = 3). Cumulative percentages of dissolved drug from preparations were calculated using calibration equations.

## 3. Results and Discussion

### 3.1. Drug Solubility Study

The first step for the design of a potential efficient formulation based on the use of cyclodextrins is to determine the impact of cyclodextrins or their derivatives on AmB solubility. A HPLC/UV method was developed and validated in terms of specificity, precision, accuracy and linearity. The analytical method proved to be linear over the concentration range of (0.5–50) µg/mL on three days (*n* = 9) in agreement with ICH guidelines (ICH-Q1-R2) [23] The chromatographic separation was obtained at 12 min. using the described method and the chromatographic parameters were summarized in Figure 1.

To complete this study, we determined the influence of cyclodextrin-based polymers (Poly-CD) and their composition on AmB solubility. Initially, Poly γ-CD, Poly β-CD and Poly-per-methyl β-CD (cyclodextrin homopolymers), Poly αβ-CD, Poly αγ-CD and Poly βγ-CD (cyclodextrin terpolymers) and Poly αβγ-CD, Poly αβγ-CD chitosan polymers (cyclodextrin tetrapolymers) were synthesized according to the organic solvent-free method reported by Skiba et al. [18]. We tested a lot of different Poly-CD, because drug interactions with such polymers could interact through inclusion in cyclodextrin cavities but also through polymer adsorption or inclusion in cavities between polymer ramifications. Thus, it is difficult to extrapolate that the best inclusion complexation with a determined native cyclodextrin leads to the best complexation with its polymerized form. From results of AmB solubility studies, Poly γ-CD showed the best enhancement with an increase of AmB solubility multiplied by a factor 127 to reach 761 µg/mL. An important interaction was also observed between polyene and terpolymers (Poly αγ-CD and Poly βγ-CD) increasing the solubility to 463 µg/mL and 461 µg/mL, respectively. Only with polymers containing γ-CD in their composition as Poly αβγ-CD, a minor improvement of AmB solubility was remarked, while no improvement was shown using other Poly-CD: Poly αβ-CD, Poly α-CD, Poly β-CD or poly per-Me β-CD. The solubility of AmB molecules was significantly increased only when γ-CD were mostly used as complexing agent. Poly-CD are high hydrophile polymers without hydrophobic block [19]. There are, thus, no amphiphile substance such as PEG-PLGA allowing the formation of micelles to improve the solubility of AmB [24]. The core of CD has hydrophobic properties but the outside of CD is high hydrophilic due to their -OH functions. This explains why these results demonstrate that increasing solubility was a consequence of more interactions through hydrogen bonds leading to inclusion complexation in cyclodextrin cavity. An easy substitution of water molecules by AmB is due to the polar–apolar energy through hydrogen bonds or Van der Waals interactions and to the reduction of intermolecular interactions responsible for low solubility. This could also be due to the esterification of AmB hydroxyl, amine or carboxyl group by the presence of citric acid within the polymeric structure [17].

### 3.2. Characterization of Designed Formulations

Several characterizations of the different designed formulations were performed and compared to AmB free form in order to better understand the influence of formulation on physico-chemical properties and interactions between γ-CD (or derivatives) and AmB. 

#### 3.2.1. Amphotericin B Content

The quantification of AmB within the different formulations using chromatographic validated method was given in the Table 1, as well as the mean diameter of powders. All the formulations prepared by spray-drying were found to be monodisperse. Drug content ranged from 90.0% to 100.0% with a homogeneous distribution of AmB in the powder (*n* = 3). Low coefficient of variance (CV) values (<1.0%) in drug content indicates the reproducibility of the technique employed for the preparation of solid dispersion (SD). The spray-dried powders were produced at different yields where the mass quantities found were higher than 45%.

#### 3.2.2. Morphology and Particle Size Analysis

The results are summarized in Table 1. The particles of free form AmB have an average size in the micrometer range, 6–8 μm. By comparison, the average size of AmB particles obtained from a spray-dried water–methanol solution was between 8 and 26 microns. This size difference could be due to the limited solubility of AmB in methanol/water mixture, resulting in agglomeration of AmB molecules. Size comparison between the different formulations containing CDs polymers shows that the formulations based on γ-CD homopolymers have a smaller average size that can enhance their wettability. PDI values >1 indicate a polynomial population of particles which can be explained by the presence of free amphotericin B in micelle form. 

In Figure 2 shows SEM micrographs of spherical amorphous solid dispersions, revealed clear changes in the morphology of powder particles with evident formation of spherical amorphous solid dispersions.

#### 3.2.3. Differential Scanning Calorimetry Analysis

The Differential Scanning Calorimetry (DSC) curve of free form AmB showed a sharp endothermic peak at *T* = 174 °C corresponding to its melting point and suggesting its crystalline nature, followed by a glass transition temperature (*T*_g_) at *T* = 200 °C [25]. However, in the physical mixture, a double pic can be observed: one corresponding to the polymer and one indicating a presence of crystalline form of amphotericin B. This second pic is displaced towards a lower temperature and the broadening can due to a weak interaction. Concerning the thermal comportment of AmB, a new peak of solid dispersion appeared corresponding to a new entity formed as a consequence of the interaction between AmB and γ-CD polymeric network and cavities (Figure 3).

The disappearance of the AmB endothermic peak was a strong evidence of an amorphous transformation that is favorable to rapid bioavailability during in vivo drug release.

#### 3.2.4. Thermal Analysis

The thermal analysis of solid dispersion is shown in Figure 4. AmB showed critical weight loss that is stabilized around 600 °C. This is probably related to the decomposition of AMB structure. The lost weight indicates the loss of water molecules included in CD cavities. Poly γ-CD presented three stages of weight loss. The weight loss percent for amphotericin B (AmB), physical mixture and solid dispersions (SD) based AmB and γ-cyclodextrin polymers (Poly γ-CD) were given in Table 2. Previous studies have reported the presence of water molecules, depending on the relative humidity in CDs [26]. For the spherical amorphous solid dispersion, the thermogram was different from the pure materials. This demonstrated the occurrence of interactions between the drug and carriers leading to the formation of new system. This present study also showed that the dehydration pattern of solid dispersion was different from the thermal behavior of pure CD and physical mixture. The percentages of water evaporation decreased when the complex was formed, this fact could be explained by the substitution of water molecules by AmB in one hand, and by the solid dispersion method itself that eliminate host water molecules, in other hand.

#### 3.2.5. Infrared and Raman Spectroscopy Property

The spectra of active ingredient showed the presence of a characteristic peak (Table 3) [27]. A broad absorption band connected to a strong bond corresponding to a ν_OH_ stretching vibration that explains the presence of intermolecular hydrogen bonds between molecules of polyene [24].

The interactions between γ-CD and AmB were studied by infrared spectroscopy. The elongation of the absorption band at 1020 cm^−1^ characteristic to the β-glycosidic linkage and the disappearance of the absorption bands at 1396 cm^−1^ could translate the interaction of AmB with the hydrophobic cavity of γ-CD via a C–H bond. An increase in intensity and a slight shift of the absorption band between 3000 and 3500 cm^−1^ was observed and that could be explained by the host–guest interactions as a consequence of water release upon inclusion. The disappearance of characteristic bands of AmB (N–H, and C=O) means the formation of single compound from AmB and γ-cyclodextrin as a result of AmB inclusion in γ-CD characterized by a sufficient cavity size (approximately 8.5 Å) for receiving an antifungal molecule (Figure 5).

About cyclodextrin polymers used, we noted a broad absorption band at 3330 cm^−1^ that corresponds to the stretching vibration ν_OH_ and explains a high polymer molecular association through intermolecular bonds between dimer entities. According to Skiba et al. [18], this interaction is due to condensation reaction between C=O carboxylic of citric acid and hydroxyl group (–OH) from the oligosaccharide entity (natural CD). The spectral analysis and comparison between solid dispersion-based homopolymer CDs and physical mixture (Figure 5) were illustrated by the higher frequency shift of the unsaturated acid carbonyl C=O stretching band from 1690 cm^−1^ at 1719 cm^−1^ in the spray-dried formulations, when it was maintained in the physical mixture. This change was caused by decreasing the crystallinity of AmB molecules present in the polymer matrix [25].

The spectral translation of the solid dispersion-based Poly γ-CD was essentially manifested by a great reduction or disappearance of the band at 3360.5 cm^−1^ corresponding to the OH stretching vibration and strong hydrogen bonds [OH…HO], that can be explained by the dissociation of intermolecular hydrogen bonds between the molecules of the AmB, and suggesting a monomeric AmB dispersion in the matrix. This phenomenon was absent in the inclusion complex with γ-CD. Another band was disappeared at 2930 cm^−1^ (CH_alkyl_) in the SD when this loss was not remarked in the IC containing Is the space necessary? CD suggesting a difference of AmB interaction between the γ-CD and the poly γ-CD.

The Raman spectroscopy analysis of pure AmB presents three bands at 1001.8 cm^−1^, 1155 cm^−1^ and 1559 cm^−1^, corresponding to C–C–H, C=O and C=C, respectively (Figure 5) [9,10]. Principally, the comparison between the Raman spectra of amphotericin B, physical mixture (PM) and solid dispersion (SD) were distinguished by a modification at spectral regions concerning polyenic vibrations (Figure 5). A first major change was reflected by an increase in the peak intensity characteristic to AmB C–CH bonds in the SD and IC that could be explained by the inclusion of AmB in the solid matrix [28]. This interaction was not observed in the physical mixture. The shift to lower frequency by a wavenumber of 3 cm^−1^ of the principal band from 1559 cm^−1^ to 1556 cm^−1^ was resulted to the delocalization of π electron system caused by the broadening from the 1535 to 1556 cm^−1^, indicating an AmB crystallinity decrease. The transformation was approved by the strong decrease of the band at 1008 cm^−1^ of infrared spectroscopy (Figure 5).

Those changes reflect the amorphous transformation of AmB molecules in the polymeric dispersion [29]. The differentiation between the inclusion complex, the solid dispersion based on γ-CD entities and γ-CDs homopolymer was mainly expressed by the band widening between 1535 cm^−1^ to 1556 cm^−1^ in the solid dispersion meaning a transformation of crystalline AmB to amorphous state while it did not appear in the inclusion complex spectra analysis (Figure 6). Another important shift to a lower frequency from 1559 cm^−1^ to 1556 cm^−1^ in both formulas means that AmB was not included only in the cavities of the γ-CD but also dispersed in polymeric networks [30].

#### 3.2.6. Aggregation Studies by UV

The state of AmB molecule aggregation within the solid dispersion was studied in order to obtain more information on the aggregation state present in the developed formulation for a possible prediction of their efficacy and toxicity [15]. The spectrophotometry of AmB absorbance may reveal both the AmB molecular form present in the aqueous medium and the conformational state of such molecules. Those conformations are responsible of efficacy and toxicity. In addition, the influence of other constituents on different AmB aggregation states has been studied to found the more stable form [31]. Free amphotericin B cannot be completely dissolved in water during our aggregation studies. Methanol was used to obtain a solvent (MetOH:water 1:1) able to dissolve the free form of this drug. This solvent was the same for all studied formulations and, therefore, induced the same potential alteration of drug interaction with polymer. 

The UV spectrum of AmB in MeOH/H_2_O mixture at approximately 10 µg/mL was principally represented by the presence of four bands, one at 347 nm followed by three well-separated bands at 364 nm, 384 nm and 408 nm (Figure 7). The last one was considered as the principal absorption confirming the great presence of AmB monomer form and different aggregate forms [32,33]. In the literature, the spectrum of Fungizone^®^ showed an intense broad peak at 329 nm and a lower absorption for the other peaks resulting in the self-aggregation of AmB [16]. The increase of AmB concentration was revealed by the improvement of the first band at 347 nm and the apparition of another band at 421 nm suggesting the high aggregation of AmB in aqueous solution [32]. 

The UV spectrum of AmB in MeOH/H_2_O mixture at approximately 10 µg/mL was principally represented by the presence of four bands, one at 347 nm, followed by three well-separated bands at 364 nm, 384 nm and 408 nm (Figure 7).

The analysis of the obtained spectrum can be used to determine a ratio between Peak I and peak IV considered as an index of relative aggregation state. The index of AmB and Fungizone^®^ was equal at 0.28 and 3.7, respectively.

The electronic absorbance of amorphous Poly γ-CD solid dispersion was illustrated by the presence of the three bands at 365 nm, 384 nm and 408 nm showing the great presence of monomeric AmB [4]. The lack of those bands (329 nm and 421 nm) can indicate that the AmB has a good monomeric distribution within the polymeric network, was generally in monomer form and may present also in the dimeric form so the most preferred aggregation state in terms of drug action and toxicity [16,32]. In aqueous medium, the dimerization by the association of AmB monomers was quickly established before the aggregation in water. The dimers in their different assemblage (parallel and anti-parallel) were stabilized by Van der Walls interactions also hydrophobic interactions responsible for aggregation [33,34,35]. The dimerization and aggregation phenomena were prevented by Poly γ-CD and the AmB monomers present in the mixture. The distribution of AmB in aqueous solutions at over a critical concentration (at C > 10^−7^ M) was distinguished by the presence of three different shape balances. These forms were dependent to the AmB concentration, the medium of solubilization and the action of the solid matrix. The highest toxicity was caused by the aggregates, less toxicity was obtained by the oligomer, essentially a dimeric form while the best antifungal active form was the AmB monomer that reduces the renal toxicity [32], demonstrating the utility to design AmB in solid dispersion with Poly γ-CD as matrix.

### 3.3. In Vitro Dissolution Study

The dissolution study compared to drug in free form is one of the crucial tests to obtain information about predicted in vivo behavior of designed formulations and their contained drug. The dissolution studies of AmB alone, and AmB included in solid dispersion at simulated gastric fluid (pH = 1.2) was remarked by the diminution of present quantity in the medium due to the degradation caused by the acid catalysis [36]. While the dissolution of AmB from the SD at pH = 7.4 was completely done after 45 min, it was not valid for the pure drug where only 60% was entirely dissolved (Figure 8).

At a low surfactant concentration (0.02% *w*/*v*), the dissolution profile of the SD at high AmB loading (approximately 15%) was rapid and 30-times greater than the AmB alone after 120 min. At a higher surfactant concentration (0.1% *w*/*v*), a rapid release profile was obtained for SD-based homopolymer-CD dispersions with complete dissolution at 40 min while at the same moment, only 40% of AmB in free form was dissolved (Figure 9). Increasing the amount of methanol in the initial composition of the formulations induced larger particles, leading to both a reduce wettability and delayed dissolution profiles. 

The comparison of the dissolution curves of γ-CD and Poly γ-CD entities gives crucial information about the AmB distribution in these two different matrices. A complete release achieved within 40 min was observed for the solid dispersions that proves the presence of monomeric dispersion of the AmB. This was the consequence of more interactions between AmB and Poly γ-CD, probably due to H-binding with acid functions of citric acid. With γ-CD, only drug interactions with the hydrophobic cavities of cyclodextrins could be performed, limiting the number of host molecules in this matrix. Hence, the presence of an excess of AmB in an aqueous medium gives rise to the dimerization of the polyenes between them through intermolecular interactions, preventing dissolution. This explains why only 65% of the AmB were dissolved even after 2 h with inclusion complexes formulation.

For the same AmB amount loading (approximately 15%), the difference between the IC (γ-CD: AmB) and ASD (based Poly γ- or Poly αγ-CD) was clearly observed (Figure 10). At 20 min, only half of the present AmB quantity was dissolved from the IC compared to the ASD. Additionally, the AmB from the different ASD was completely dissolved after 45 min. This is due to the good exposition of AmB particles to the dissolution medium because of the difference in drug loading and the fact that the CD polymers allow a good wettability of AmB particles which are well dispersed in the system. So, the polymeric structure offers to the antifungal drug the possibility of state change and decreasing crystallinity, preventing the polyenic molecular association and permitting a good dissolution due to good exposition particle size reduction. These results could be explained by a better dispersion of AmB monomers in the polymeric structure.

The physical mixture demonstrated a slight interaction between AmB and CD polymers by DSC that was confirmed by the significance of spray-drying process in the dissolution comportment in terms of the elimination of residual humidity and amorphous change state (Figure 10). The importance of the atomization at high pressure and temperature was confirmed by the rapid dissolution spray-dried AmB (56% reached at 15 min compared to 30% for AmB bulk).

The dissolution behavior of solid dispersions was impacted by the choice of polymer compound and its composition in natural cyclodextrin, the ratio of AmB in the powder and spray-drying processing conditions. Even the slight difference in terms of solubility between γ-CD and the polymer of γ-CD, the advantages of polymeric structure were observed in dissolution studies, and AmB onomeric structure within polymers in the solid state. The fundamental properties of CD polymers were expressed in a different behavior in the solid state. An amorphous state is obtained by the spray-drying technique where during atomization, the H_2_O molecules were discarded and AmB molecules were placed in the dense polymeric network. The result was confirmed by the inability to charge the γ-CD in the solid state even after atomization. In spite of the strongest Van der walls interactions in the solid state because the shortness of distance, γ-CD polymers have a capacity of offering a good AmB distribution to decrease the aggregation phenomena.

## 4. Conclusions

The concept of spherical amorphous solid dispersions 4.0 brings an interesting formulation alternative with the use of innovative Poly-CD. Indeed, this allows us to develop powders having high AmB monomers loading (15% (*w*/*w*)) in comparison to inclusion complex with natural cyclodextrin. The solid particles were advantageous for the improvement of the dissolution profile (100% at 40 min at pH = 7.4) through the solubility increase, as well as the reduction of the particle size allowing good wettability of particles in dissolution. Also, the controlled release can be obtained by this 4th generation of solid dispersions [17]. Furthermore, the synthesis process of different polymers of cyclodextrins as described by Skiba et al. [18] is simple, rapid and without organic solvent, and their biodegradable composition may offer a human use possibility. The physicochemical properties and the behavior of AmB in the polymeric matrix were demonstrated by spectroscopy. The infrared spectroscopy gives us an idea about the interaction between the polyene and oligosaccharide of cyclodextrin. The inclusion of AmB within the cavities was also confirmed. This was confirmed by both, (i) the low waters quantities present in the spray-dried formulations as a consequence of water molecules substitution in the inner cavity of CD, and (ii) the decrease with Poly-CD of AmB crystallinity observed by DSC, Raman and infrared spectroscopy. The amphotericin B amorphization was probably a consequence of its interaction with Poly-γ-CD by dipole–dipole interactions and without a creation of covalent binding with the host molecule because the drug release was facilitated by the binary matrix. The spray-dried amphotericin B amorphous solid dispersions 4.0 technology showed, in consequence, the possibility of drug development for oral utilization or as powder aerosols for pulmonary administration. 

## Figures and Tables

**Figure 1 pharmaceutics-10-00235-f001:**
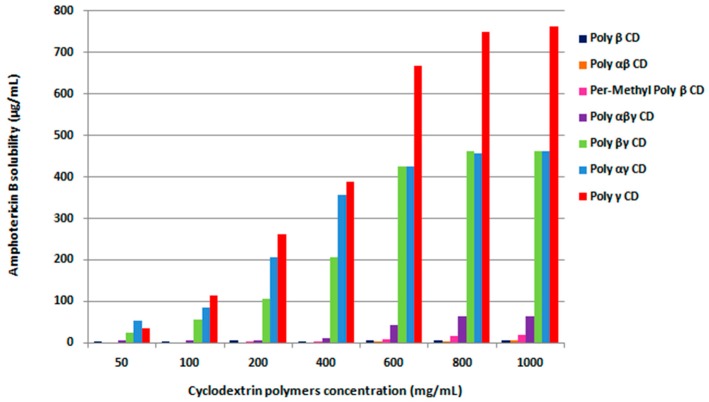
Solubility diagrams of AmB in presence of cyclodextrin polymers at 37 °C (*n* = 3).

**Figure 2 pharmaceutics-10-00235-f002:**
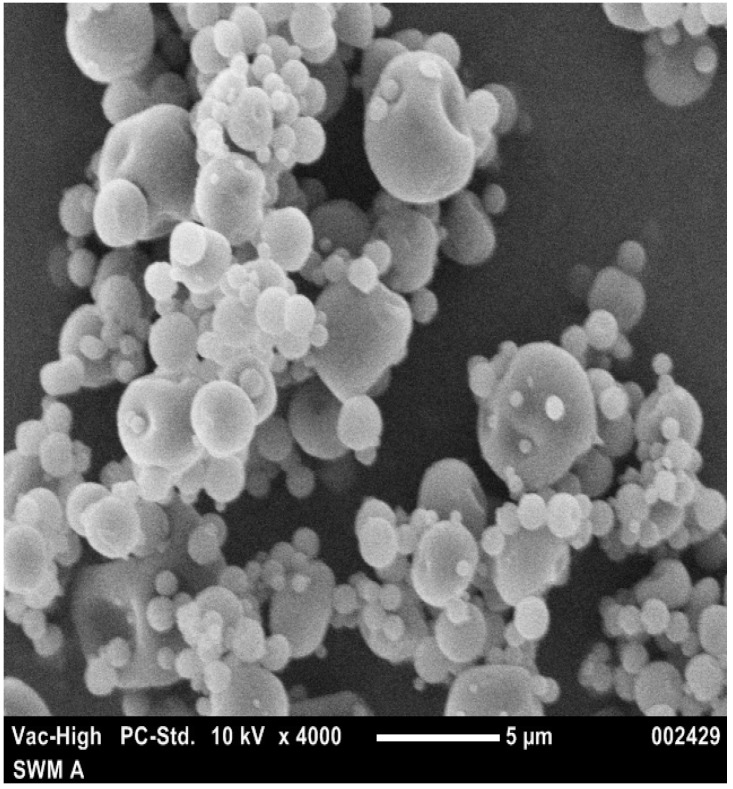
SEM photomicrograph of AmB spherical amorphous solid dispersions.

**Figure 3 pharmaceutics-10-00235-f003:**
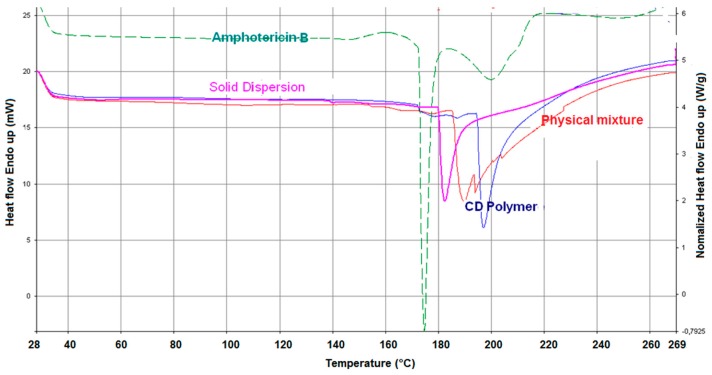
Thermograms of spray-dried amphotericin B (green), uncharged γ-cyclodextrin based polymer (blue), solid dispersion with AmB (purple) and physical mixture of γ-cyclodextrin based polymer with AmB (red).

**Figure 4 pharmaceutics-10-00235-f004:**
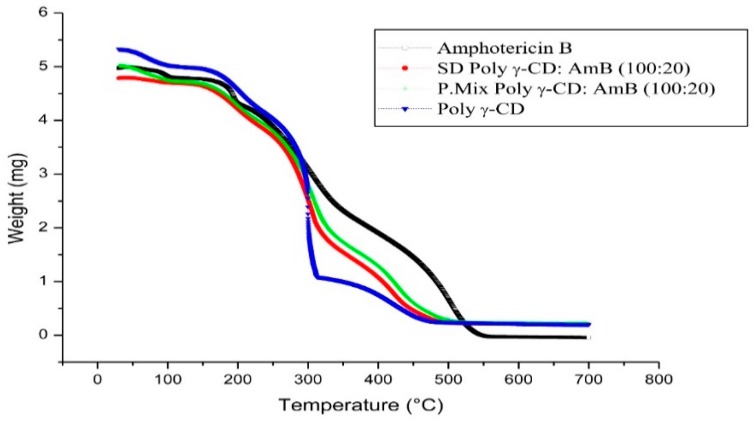
Thermal analysis of amphotericin B (AmB), uncharged solid dispersion of γ-cyclodextrin based polymer (Poly γ-CD), physical mixture of γ-cyclodextrin based polymer with AmB and solid dispersion based Poly γ-CD and AmB (SD Poly γ-CD/AmB).

**Figure 5 pharmaceutics-10-00235-f005:**
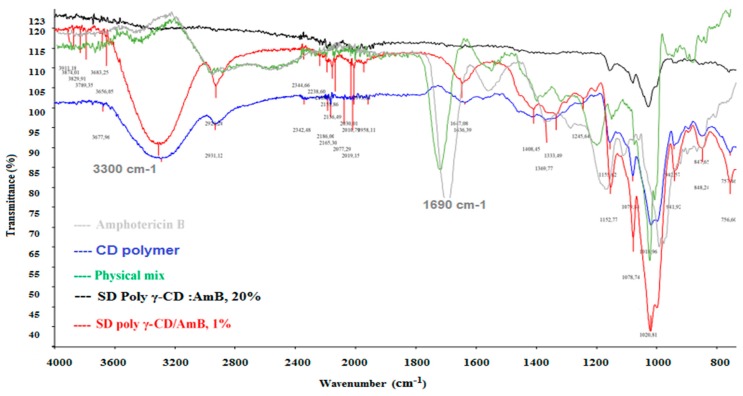
FTIR spectra of amphotericin B, uncharged solid dispersion of γ-cyclodextrin polymer (poly γ-CD), physical mixture and amorphous solid dispersion based γ-cyclodextrin polymer and AmB (SD poly γ-CD/AmB).

**Figure 6 pharmaceutics-10-00235-f006:**
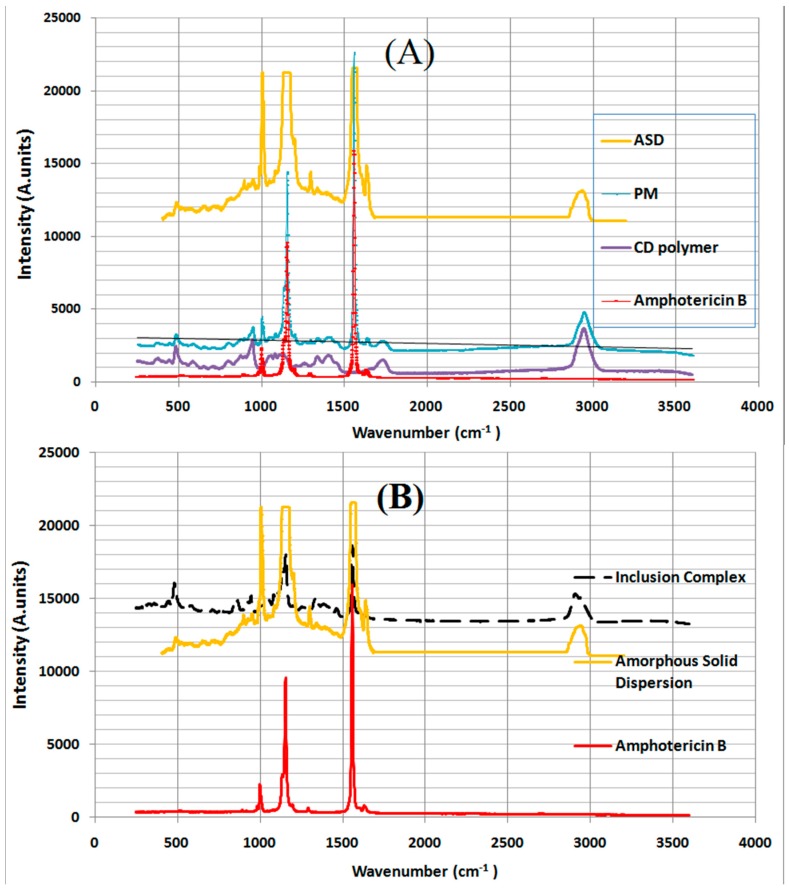
(**A**) Raman spectra of amphotericin B, cyclodextrin polymer (Poly-γ-CD), physical mixture (PM) and amorphous solid dispersion (ASD). (**B**) Raman spectra of amphotericin B, ASD and inclusion complex.

**Figure 7 pharmaceutics-10-00235-f007:**
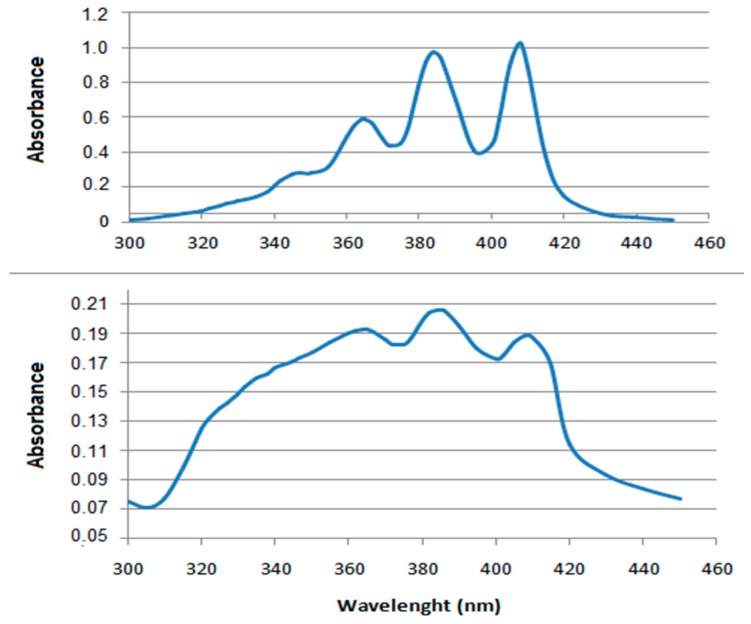
Spectra of electronic absorbance of AmB in MeOH:H_2_O (1:1) at 10 µg/mL (**top**) and solid dispersions of γ-cyclodextrin polymers and amphotericin B (100:20) at 10 µg/mL in deionized water (**bottom**).

**Figure 8 pharmaceutics-10-00235-f008:**
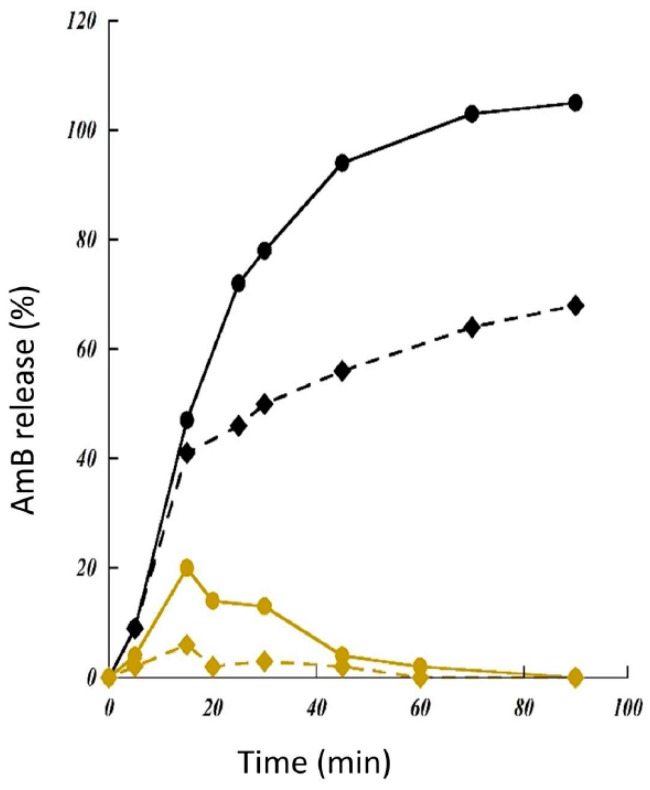
In vitro release profiles of amphotericin B (AmB) in gastric (gold, pH = 1.2) or pulmonary media (black, pH = 7.4). Drug was in free form (lozenges) or contained in solid dispersions based γ-cyclodextrin polymers (filled circles).

**Figure 9 pharmaceutics-10-00235-f009:**
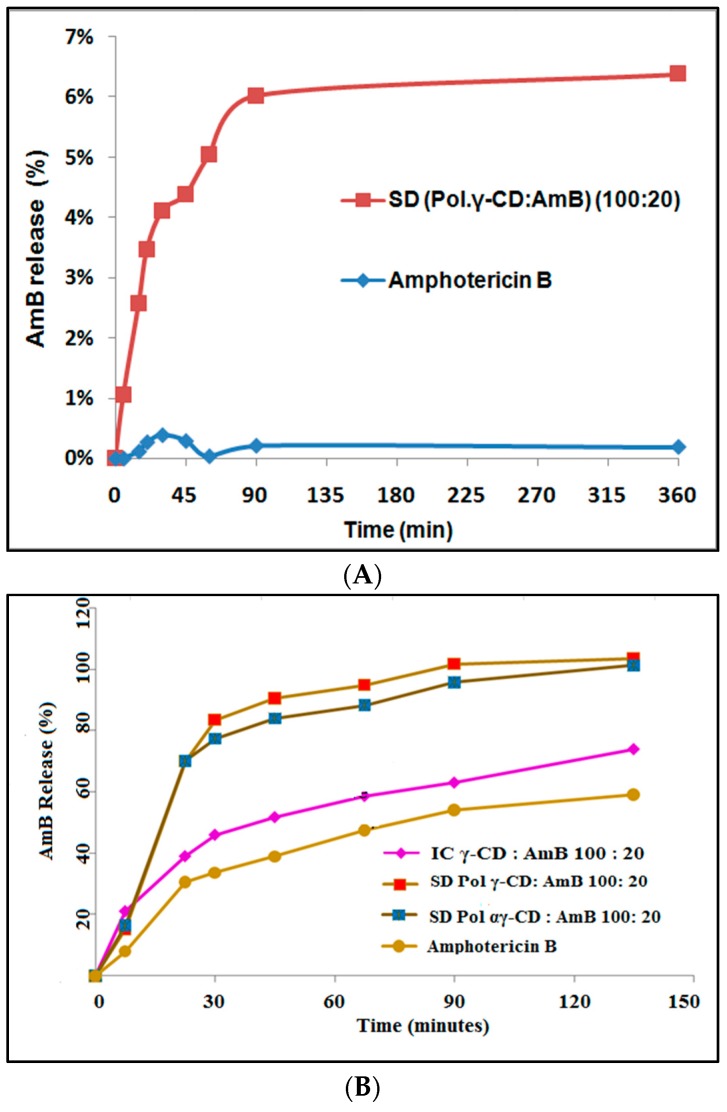
Dissolution profiles of AmB, solid dispersions (SD) with different polymers composition (αγ-poly CD, γ-poly CD) and inclusion complex (IC) in simulated fluid medium pH = 7.4 with 0.02% (**A**) or 0.1% SLS (**B**).

**Figure 10 pharmaceutics-10-00235-f010:**
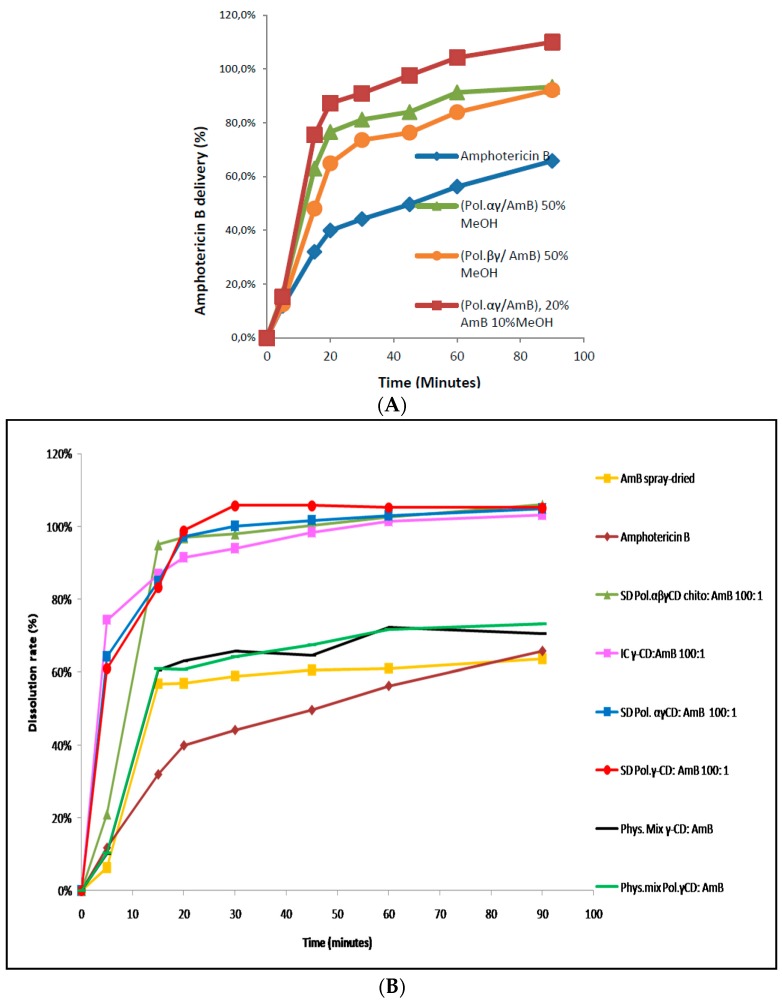
Dissolution profiles of cyclodextrin terpolymers (βγ-CD, αγ-CD) for different methods of preparation (**A**). Comparison of dissolution profiles between the different spray-dried formulations solid dispersions (SD), AmB: bulk and spray-dried, physical mixture (Phys. Mix) (**B**).

**Table 1 pharmaceutics-10-00235-t001:** Dosage (%) and mean diameter particles of amphotericin B drug (AmB), and different spray-dried formulations: solid dispersions (SD), γ-cyclodextrin based polymers (Poly γ-CD) and αγ-cyclodextrin based polymers (Poly αγ-CD). 95% limits mean the limits of size of 95% of the particle populations.

Formulations/(%) MeOH	Dosage (%)	Mean Diameter (nm)	Sd (nm)	95% Limits (nm)	Indice of Polydispersity
Amphotericin B	-	7090	370	5560–8630	1.6
AmB spray-dried (sd)/10%	-	11,200	605	8130–14200	1.4
AmB spray-dried (sd)/50%MeOH	-	19,500	643	12,500–2,6500	0.8
SD (Poly γ-CD/AmB) (100:20)/10%	13.72%	1170	46	1060–1270	1.2
SD (Poly γ-CD/AmB) (100:10)/10%	5.19%	688	24	641–734	1
SD (Poly γ-CD/AmB) (100:10)/30%	5.39%	1820	52	1620–2020	1.1
SD (Poly αγ-CD/AmB) (100:20)/10%	13.17%	3620	106	3060–4190	1.2
SD (Poly αγ-CD/AmB) (100:1)/10%	5.27%	1550	47	1400–1710	1.2
SD (Poly αγ-CD/AmB) (100:10)/50%	8.37%	5640	221	4550–6730	1.2
SD (Poly γ-CD/AmB) (100:10)/50%	8.36%	5450	218	4420–6490	1.2

**Table 2 pharmaceutics-10-00235-t002:** Weight loss in percent of amphotericin B (AmB), physical mixture and solid dispersions (SD) based AmB and γ-cyclodextrin polymers (Poly γ-CD) calculated from the dehydration step of thermal analysis.

Analyzed Products	Dehydration Step		
	Temperature Range (°C)		Weight Loss (%)
Amphotericin B	30	105	3.40
Poly γ-CD	30	105	5.69
Physical Mixture	30	105	5.60
SD [Poly γ-CD/AmB]	30	105	1.87

**Table 3 pharmaceutics-10-00235-t003:** Characteristic bands of amphotericin B in infrared spectroscopy.

Compound	Absorption (cm^−1^)	Nature of Bond	Type of Vibration
Amphotericin B	1557.8 cm^−1^	δ_s_N–H (in the plan)	Deformation
	1590 cm^−1^	ν_C–C_	Stretching
	1690.17 cm^−1^	ν_asC=O [COOH]_	Stretching
	2934.5 cm^−1^	ν_s+as [CH2, CH3]_	Stretching
	3360.6 cm^−1^	ν_[CH] polyene_	Stretching

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
