# Peer review of "Preparation and Characterization of Spherical Amorphous Solid Dispersion with Amphotericin B"

_pharmaceutics, 2018, doi:10.3390/pharmaceutics10040235_

Round 1
Reviewer 1 Report
The manuscript by Mehenni et al. describes efforts to improve the solubility and dissolution of amphotericin B by mixing it with cyclodextrin polymers. A variety of characterization techniques was carried out to analyze the resulted products such as particle size, particle distribution, chemical structure, etc. Finally, the release of amphotericin B from such formulation was investigated. The results are interesting and could be an interest to broad readers. I suggest the following comments to be addressed prior to be considered for publication:
- Table 1 is so confusing. There are many terms that require clarification hence add the information into the text. How to calculate "dosage"? How come polydispersity index has value of more than 1? Definition and how to get the polydispersity index should be added. Also, what is the meaning of 95% limits? Please add the explanation into the text.
- I don't understand why the drug content percentage is more than 100% (line 234).
- What is A in the "A.units" on y-axis title on Figure 5? Explain abbreviation on Figures.
- In terms of the release, it is not clear what "cumulative release" means in Figure 7's y-axis.
General comments:
- Tables 1 and 3: the use of comma is often interchanged with period, e.g. "13,72%" where it should be 13.72%. Please check the entire manuscript and change as necessary.
- In Table 3, the font size is not homogenous. Please revise.
- In Table 3, the unit cm-1 is redundant as it has been provided in the column title. So, delete the ones in the column content.
- In Figures 1, 5, 8, and 9A remove ",0", ",0" ,",00" and ",0" respectively from the y-axis
- In Figure 3, remove "Top"
- Revise Figure 4 to higher resolution
- In Figure 6, revise the x-axis title from "wavelenght" to "wavelength". Please check and revise typos in the entire manuscript.
- In Figure 8, the x-axis titles are not consistent. Please revise.
Author Response
Please find a point by point response to your questions
thak you for your help
best regards
M SKIBA

Reviewer 2 Report
The manuscript entitled “Preparation and characterization of Spherical Amorphous Solid Dispersion of amphotericin B” is devoted to the drug delivery improvement. In order to achieve this aim, spray-drying technique and cyclodextrin polymers were used. In my opinion the manuscript will be very interesting for researchers dealing with drug form development. The obtained solid dispersions were comprehensively analyzed using different methods including SEM, DSC, FT-IR and TGA. The results were clearly presented and well organized. Undoubtedly, this work is innovative and valuable. However I have some suggestions/comments which are listed below:
Introduction.
The introduction is very interesting, however, there is no information about spray-drying and similar techniques. I suggest to mention in the introduction about methods utilizing solvent evaporation in drug form development. In the recent years several interesting examples of solvent evaporation methods appeared including droplet evaporative crystallization (salicylic acid, aspirin, benzoic acid), double emulsion solvent evaporation (Fucoxanthin), ultrasound and solvent diffusion-evaporation (Curcumin). Then, the authors could mention about recent studies on the spray-drying (Diazepam, Repaglinide).
Results and discussion
Line 290
I suggest to use “red-shift” and “blue-shift” terms, which give concise information about IR bands shifts.
Line 392
How many repetitions of experiments were performed? I suggest to provide error bars on the plots or information about standard deviations in the text.
In my opinion, the language should be improved. I found some errors listed below:
There is a lack of space in several sentences
„in 1970 [1].In medicine” (line 29), „permeability [3].This” (line 33), „0.2mg/ml” (line 34), “isolated and quantifiedusing an” (line 77), “the study at anaccelerated voltage” (line 169), “energy through hydrogenbonds” (line 223)
Non-English words or phrases:
Line 73
“Lauryl Sulfate de Sodium”
Table 3
polyène
Other minor language errors:
Line 36
Please change “using a ultra-violet [UV] spectrophotometer” to “using an ultra-violet [UV] spectrophotometer”
Line 50
Please change ” CD based polymers [Poly-CD]” to “CD-based polymers [Poly-CD]”
Line 98
Please change “The yellow orange suspensions” to ”The yellow-orange suspensions”
Line 157
Please change “noise ratio were obtained with integration times” to “noise ratio was obtained with integration times”
Line 205
Please change “influence of cyclodextrin based polymers” to “influence of cyclodextrin-based polymers”
Line 258
Please change “However, a largest and” to “However, the largest and”
Line 261
Please change “a consequence of interaction between” to “a consequence of the interaction between”
Line 310
Please change “hydroxyl groupment” to “hydroxyl group”
Lines 315, 460
Please change “cristallinity" to”crystallinity”
Lines 324, 460, 290
Please change “raman” to Raman”
Line 358
Please change “peaks resulting to the self-aggregation of” to “peaks resulting in the self-aggregation of”
Line 359
Please, “revealed by the improve of the first band at” to “revealed by the improvement of the first band at”
Lines 347, 385, 403,
Please change “informations” to “ information” (information is uncountable noun)
Line 406
Please change “probable due to H-binding with” to “probably due to H-binding with”
Line 452
Please change “Furthermore, the synthesize process of different” to “Furthermore, the synthesis of different”
Line 459
Please change “substitution in inner cavity of” to “substitution in the inner cavity of”
Author Response
Please find a point -by-point response for your question
thank you for your help
best regards
M SKIBA

Reviewer 3 Report
Current manuscript aims to prepare a solid dispersion for poorly aqueous soluble amphotericin B using cyclodextrin as a carrier. Different formulations were prepared and characterised for their physicochemical properties. Solid dispersion technology is well-known for its ability to improve the drug solubility. In addition, use of cyclodextrin as a carrier to enhance solubility have been used for many decades. In terms of technology advancement in drug delivery, this manuscript lacks use of any novel ideas. This work is rather routine and may not contribute much in advancement of drug delivery. Although, the formulation optimisation aspect in the manuscript was nicely characterised, the manuscript lacks major experimental studies to prove superiority of using the authors acclaimed formulation. I would strongly suggest authors to do additional studies such as invitro antifungal and in vivo pharmacokinetics .
There are several typos and grammar issues in the manuscript. Therefore, it requires scientific proofreading and rewriting where necessary. Authors acclaimed this formulation can be used by pulmonary route however the release studies are conducted in simulated gastric fluid media. Authors need to make sure which route of administration the formulation would be used for as physicochemical properties of particles or formulations play significant role in selection of intended therapy.
Based on my comments above, novelty of ideas and lack of experiments (in vitro and in vivo) to justify superiority of the formulations over existing one, I am rejecting the paper for publication in highly impact journal like pharmaceutics.
Author Response
Please find a point-by-point response for your questions
Thank you for your help
best regards
M SKIBA

Round 2
Reviewer 3 Report
Looks fine after modification. It can be accepted now.